# Integrative genome-wide analysis reveals the role of WIP proteins in inhibition of growth and development

Maria Victoria Gomez Roldan [1,3], Farhaj Izhaq[1,3], Marion Verdenaud [1,2], John Eleblu [1], Aimen Haraghi[1], Vivien Sommard[1], Pierre Chambrier[2], David Latrasse[1], Teddy Jégu[1], Moussa Benhamed[1], Judit Szécsi[2], Mohammed Bendahmane [2], Adnane Boualem[1] & Abdelhafid Bendahmane [1✉]

In cucurbits, *CmWIP1* is a master gene controlling sex determination. To bring new insight in the function of *CmWIP1*, we investigated two *Arabidopsis* WIP transcription factors, AtWIP1/ TT1 and AtWIP2/NTT. Using an inducible system we showed that WIPs are powerful inhibitor of growth and inducer of cell death. Using ChIP-seq and RNA-seq we revealed that most of the up-regulated genes bound by WIPs display a W-box motif, associated with stress signaling. In contrast, the down-regulated genes contain a GAGA motif, a known target of polycomb repressive complex. To validate the role of WIP proteins in inhibition of growth, we expressed AtWIP1/TT1 in carpel primordia and obtained male flowers, mimicking CmWIP1 function in melon. Using other promoters, we further demonstrated that WIPs can trigger growth arrest of both vegetative and reproductive organs. Our data supports an evolutionary conserved role of WIPs in recruiting gene networks controlling growth and adaptation to stress.

[1] Institute of Plant Sciences Paris-Saclay (IPS2), INRAE, CNRS, Université Evry, Université de Paris, Université Paris-Saclay, 91405 Orsay, France. [2] Laboratoire Reproduction et Développement des Plantes, INRAE, CNRS, Université Lyon 1-ENSL, Ecole Normale Supérieure de Lyon, 69364 Lyon Cedex 07, France. [3]These authors contributed equally: Maria Victoria Gomez Roldan, Farhaj Izhaq. ✉email: abdelhafid.bendahmane@inra.fr

Flowers are the reproductive organs of angiosperms, which lead to the production of fruits and seeds. Most angiosperms bear hermaphrodite flowers, developing both male and female organs in the same flower. For the rest of the flowering plants, the selective abortion of the female or male organ primordia of initially hermaphroditic flower meristems results in unisexual flowers[1].

The production of unisexual flowers and plants is instrumental in plant breeding, as they ease the production of F1 hybrid seeds. In *Cucurbitaceae*, sex determination is wide spread[2]. Out of 800 inspected species, 460 are monoecious and 340 are dioecious[3], which makes this plant family a practical model to investigate the molecular mechanisms controlling sex determination. In melon, three sex genes, *androecious (A)*, *andromonoecious* (M), and *gynoecious* (G), have been identified to play important roles in the development of unisexual flowers and plants. The *A* and *M* genes encode for the limiting enzymes in the ethylene biosynthesis pathway, *CmACS11* and *CmACS7*, respectively[4,5]. The *G* gene encodes a zinc finger transcription factor, *CmWIP1*[6]. Expression of *CmACS7* in carpel primordia inhibits, through an unknown non-cell-autonomous mechanism, the development of stamen primordia and leads to unisexual female flowers. *CmWIP1* is a central integrator of the transcriptional networks leading both to inhibition of carpel development and the control of the expression of the stamina inhibitor, *CmACS7*. Still the identities of the transcription networks recruited by CmWIP1 to cause organ-specific abortion are mostly unknown.

CmWIP1 belongs to subclass A1d of zinc finger transcription factors, characterized by a highly conserved C-terminal domain and two conserved motifs in the N-terminal[7,8]. In the C-terminal, there are three conserved amino acids, tryptophan (W), isoleucine (I), and proline (P) on which the protein family is named WIP, followed by four C2H2 zinc finger (ZF) motifs. WIP proteins are land plants specific transcription factor, present in one copy in the liverwort *Marchantia*[9], two copies in the moss *Physcomitrella*, three copies in the genome of the lycophyte *Selaginella moellendorffii* and six copies in most diploids (such as *Arabidopsis*, tomato, and melon) and higher copies in polyploids angiosperm. So far, little is known about the molecular function for WIP proteins and no transcriptomics analysis have been done that allows identification of the main gene targets of this family of transcription factors.

As melon is not a practical model system to investigate the molecular mechanisms underlining the function of a given transcription factor, we used *Arabidopsis thaliana* to bring new insight to transcription networks recruited by orthologues of *CmWIP1* in *Arabidopsis*. In *Arabidopsis*, there are six WIP transcription factor (C2H2 family, subclass A1d) family members[10]. *TT1/AtWIP1* (*TRANSPARENT TESTA 1*) is expressed in the seed coat endothelium, where most pro-anthocyanins (PAs) are accumulated. Loss-of-function *tt1* mutant shows yellow seeds due to a reduced PAs accumulation[8]. *NTT/AtWIP2* (*NO TRANSMITTING TRACT*) is expressed in the transmitting-tract during silique development, and *ntt* loss-of-function mutants display a half-filled fertilized silique phenotype, due to the malformation of transmitting tract in the carpel[11]. A recent study has demonstrated that *NTT*, *AtWIP4*, and *AtWIP5* are redundantly expressed in the hypophysis and are required for the distal stem cell fate within the root meristem[12]. Interestingly, the function of AtWIPs in *Arabidopsis* is partially conserved/overlapping as the expression of the coding sequence of any *AtWIP* gene under the *TT1* promoter is able to restore the *tt1* yellow seed phenotype[10]. These results suggested that all WIPs in *Arabidopsis* have highly conserved domains and may only differ in their spatial and temporal expression patterns. However, it is not clear whether *WIP* genes are functionally conserved across different plant species.

Ectopic expression of *TT1* and *NTT* under the control of the *35S CaMV* promoter showed severe phenotypic aberrations with rosette leaves deeply serrated[8,13]. Here, we show that *35S: CmWIP1* transgenic *Arabidopsis* plants have identical phenotypes, indicating a conserved growth repression role for WIP proteins. To bring new insight into how WIP proteins accomplish growth inhibition, we expressed *TT1* and *NTT* genes using a dexamethasone inducible (Dex) system and performed Chromatin ImmunoPrecipitation with sequencing (ChIP-seq) and RNA sequencing (RNA-seq) analysis. This combined analysis allowed us to identify genes that are collectively regulated by two representative WIP proteins in *Arabidopsis*. Overexpression of WIP genes activated several genes involved in programed cell death and in hormone signaling related to stress responses, such as ethylene and jasmonic acid (JA), and repressed many genes involved in organ and plant development.

To test whether the WIP-mediated inhibition of growth observed in transient expression in *Arabidopsis* could mimics CmWIP1 inhibition of carpels in melon, we expressed *TT1* under the control of a carpel specific promoter. As did CmWIP1 in melon, expression of *TT1* in carpel produced male flowers. To test whether the inhibition of growth and development is a general mechanism of WIP proteins we ectopically expressed *TT1* in two more flower organs, stamina and petals primordia, and in two vegetative tissues, trichrome and lateral roots. Our results showed that WIPs are able to inhibit both vegetative and reproductive organ growth. Our data also shows that several molecular pathways controlled by WIP proteins are conserved between plant species and their role is probably determined by their spatiotemporal expression during development. Our data also point toward the use of WIP proteins as a biotechnology tool to engineer male or female plants in other species.

## Results

**Functional annotation of *WIP* genes in *Arabidopsis* and Melon.** To better understand the molecular conservation of WIP proteins, we first examined the protein sequences and the expression profiles of WIPs in *Arabidopsis* (Supplementary Figs. 1a and 2). A phylogenetic tree analysis revealed that TT1/AtWIP1 clustered together with AtWIP3, AtWIP6, whereas NTT/AtWIP2 clustered together with AtWIP4 and AtWIP5 (Fig. 1a). To investigate the conservation of WIP genes between *Arabidopsis* and melon, we searched, by BLAST protein, the recently sequenced *Cucumis melo* L. genome[14], for sequences orthologues to AtWIPs. This resulted in the identification of six *CmWIP* genes which were further analyzed according to their gene structures. As in *Arabidopsis*, melon WIP genes are composed of two exons, a N-terminal region with two conserved motifs of unknown function (n1 and n2), and a highly conserved C-terminal region on which four zinc fingers (ZFs) and nuclear localization signals (NLSs) are located (Supplementary Fig. 1b). Alignment analysis showed that two WIP proteins from melon (MELO3C004740 and MELO3C003009) were clustered with TT1, whereas CmWIP1 and three others related proteins (MELO3C009657, MELO3C023544, and MELO3C009030) were part of the NTT clade (Fig. 1a).

To test whether WIP proteins are functionally conserved between melon and *Arabidopsis*, we first overexpressed *CmWIP1*, under the control of cauliflower mosaic virus 35S promoter (*35S*), in *Arabidopsis*. The obtained transgenic plants (*35S:CmWIP1*) strongly expressed *CmWIP1* and phenocopied plants overexpressing *TT1* (*35S:TT1*) and *NTT* (*35S:NTT*) (Supplementary Fig. 3a–c)[8,13]. *35S:CmWIP1* transgenic plants also displayed smaller rosettes, serrated leaves, dwarf plants, defective floral organs, and reduced plant fertility (Fig. 1b). Root elongation,

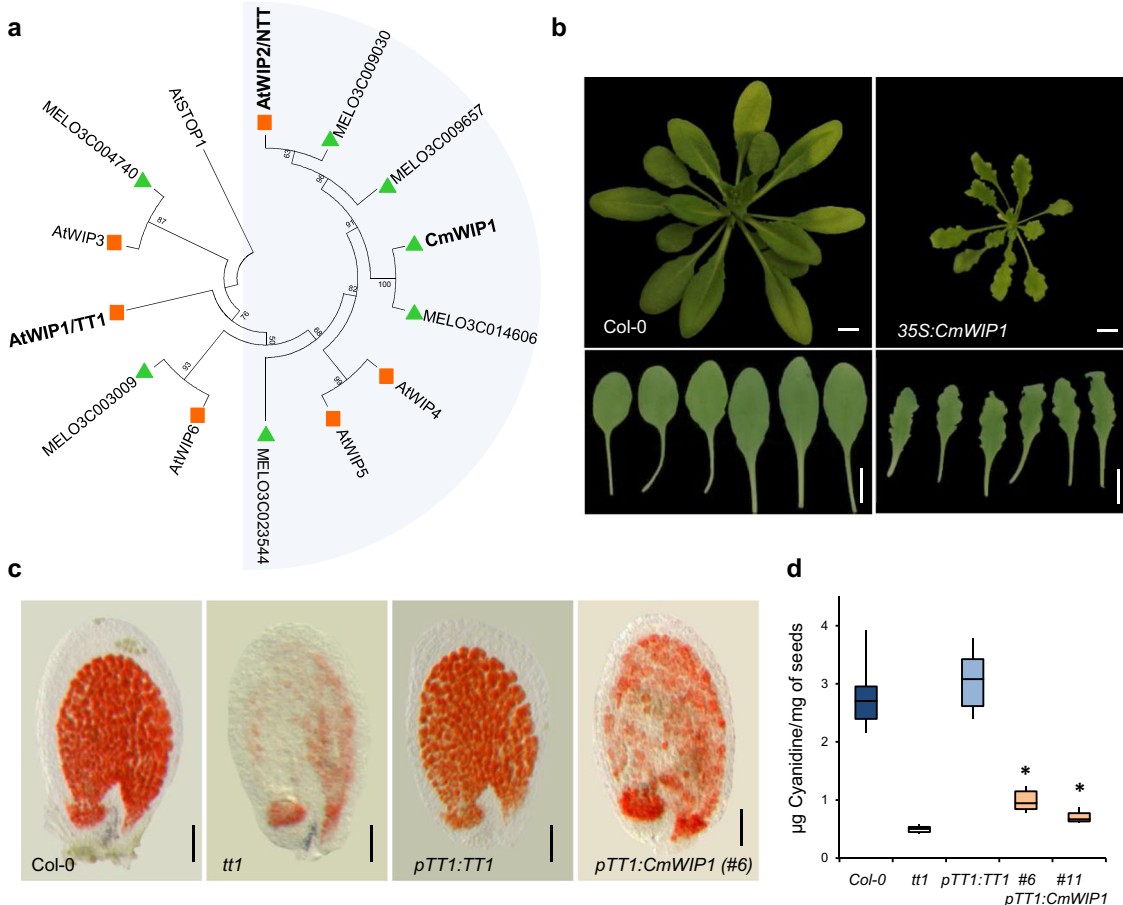

**Fig. 1 WIP transcription factors in *Arabidopsis* and Melon. a** Phylogenetic tree with the six WIP genes from *Arabidopsis* and melon. The closest homologue of CmWIP1 in *Arabidopsis* is NTT, involved in transmitting tract development in *Arabidopsis*. NTT together with WIP4 and WIP5 are required for initiation of the root meristem. **b** Rosettes and leaves of WT Col-0 and *35S:CmWIP1* transgenic lines, *35S:CmWIP1* shown serrated leaves, as previously described in *35S:TT1* and *35S:NTT*[8,13]. Scale bar = 0.5 cm. **c** Vanillin staining of *Col-0*, *tt1* mutant, and transgenic immature seeds using *TT1* and *CmWIP1* under the control of *TT1* promoter (*pTT1*) to complement yellow seeds phenotype of *tt1* mutants. Scale bar = 20 μm. **d** Quantification of pro-anthocyanidins (PA) in *Col-0*, *tt1* mutant, and transgenic mature seeds. Bars indicate standard errors of two biological repeats and three technical repeats. Asterisks show significant differences compared to *tt1* mutant (Student *t*-test, *p* < 0.05, *n* = 6 biologically independent samples, each corresponding to a pool of 10 mg mature seeds).

internode, and silique sizes were also all reduced compared to WT *Col-0* plants (Supplementary Fig. 3d–g). This similarity in the observed phenotypes suggests that common molecular processes are initiated by *CmWIP1* in *Arabidopsis* compared to its orthologues *TT1* and *NTT*.

It has been previously demonstrated that all *AtWIP* genes are able to restore pro-anthocyanins (PA) accumulation in the seed coat of *tt1* mutant by using a promoter swap approach[10]. To test whether *CmWIP1* is able to fulfill the function of AtWIPs in seed coat development, we expressed the coding sequence of *CmWIP1* under the *TT1* promoter (2 Kb) in the *tt1* mutant (Supplementary Fig. 4a). As a control, we expressed coding sequence of *TT1* under its own promoter (2 kb) in *tt1* mutant. Two independent transgenic lines that showed a high *CmWIP1* expression, were further characterized (Supplementary Fig. 4b). We analyzed the accumulation of PA in seed coat by vanillin staining in immature seeds (Fig. 1c) and cyanidin quantification from the mature seeds (Fig. 1d) for each transgenic line. Similar to the WT control, PA accumulation was fully complemented by *pTT1:TT1* construct in *tt1* mutant. Seeds from *tt1* lines complemented by the *pTT1: CmWIP1* construct showed a significant increase of PA accumulation when compared to *tt1* mutant; however WT levels were not attended. Similar complementation levels of the *tt1* lines

were obtained by Appelhagen et al.[10] when using *pTT1:NTT*. Taken together, our results demonstrate that WIP proteins share both conserved protein structure and function, between melon and *Arabidopsis*. Thus, characterization of WIP proteins in *Arabidopsis* will likely bring new insight in the function of WIP proteins in melon.

**Overexpression of At*WIPs* trigger senescence and cell death.** The cell-type-specific and transient expression pattern of *WIP* genes in Arabidopsis (*TT1* in endosperm cells of the seed and *NTT* in transmitting tract and quiescent center cells of root tips) may be a technical limitation to explore their molecular function. Furthermore, expression of WIP genes under the control of the 35S promoter led to recovery of lines with strong growth inhibitions that correlate with expression of the transgene. Inducible gene expression systems permit synchronizing target gene expression at particular developmental stages and in particular tissues. Hence, we generated transgenic lines that express *TT1* and *NTT* under the control of dexamethasone-inducible (Dex) system (Supplementary Fig. 5a, b)[15]. To test the transient expression of *TT1* and *NTT* on germination, seeds of *Dex:NTT* and *Dex:TT1* lines were sown on MS medium containing 1 μM of

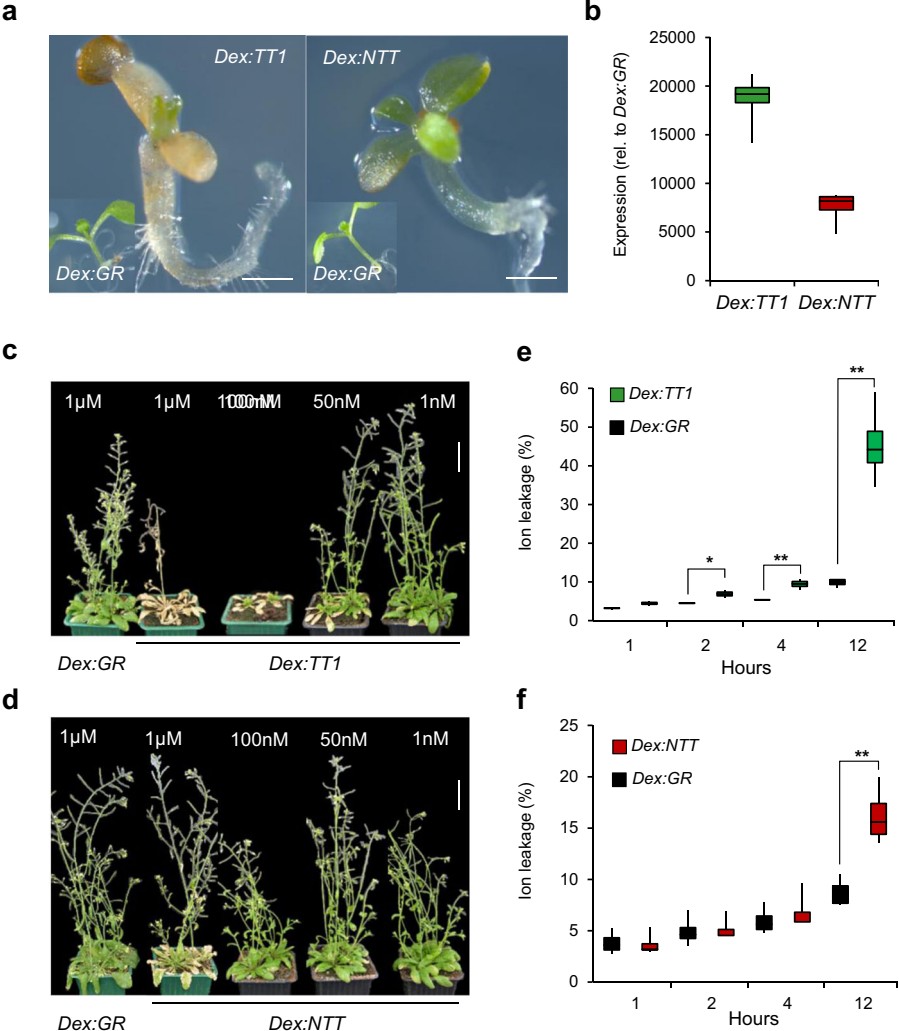

**Fig. 2 Overexpression of *TT1* and *NTT* induces growth arrest and leaf senescence. a** Phenotype of control (*Dex:GR*), *Dex:TT1*, and *Dex:NTT* transgenic lines grown on MS medium containing 1 μM of dexamethasone (Dex), scale = 0.5 mm. **b** Expression of *TT1* and *NTT* in *Dex:TT1* and *Dex:NTT* transgenic lines, respectively, relative to expression in *Dex:GR* after Dex induction (1 μM) during 8 h. Phenotype of 5-week-old *Dex:TT1* (**c**) and *Dex:NTT* (**d**) plants 4 days after spray with different concentrations of dexamethasone (1 nM to 1 μM) and compared to *Dex:GR* control plants (scale = 3.5 cm). Ion leakage measured on *Dex:TT1* (**e**) and *Dex:NTT* (**f**) at different time points on leaf disks of plants sprayed with 1 μM of Dex (compared to corresponding *Dex:GR*). Asterisks show significant differences between overexpressing lines and controls (Student *t*-test, *$p < 0.05$, **$p < 0.005$, $n = 4$, independent treated plants, from each plant three leave disks were pooled for each time point).

Dex, and compared to the control *Dex:GR* lines (only *35S:LhGR* vector). At this concentration germinated seedlings, of both *Dex:NTT* and *Dex:TT1* lines, stop growing once radicle and cotyledons emerged whereas the control *Dex:GR* line germinate and grow normally (Fig. 2a). However, in contrast to *Dex:NTT* lines, *Dex:TT1* lines displayed a more severe growth inhibition. This difference in the severity of growth inhibition, observed in all generated independent transgenic lines following Dex-treatment, could be explained by 3-fold more accumulation of *TT1* mRNA compared to *NTT* (Fig. 2b).

To investigate the effect of *TT1* and *NTT* expression on growth at later developmental stages, 5-week-old plants were sprayed with different concentrations of Dex, from 1 nM to 1 μM. Plant phenotypes were evaluated 4 days after treatment. Cell death and leaves senescence were observed in *Dex:TT1* lines even at very low concentration of Dex (50 nM) (Fig. 2c). In contrast, *Dex:NTT* plants showed leaves senescence only at 1 μM Dex treatment (Fig. 2d). No phenotypic changes were observed on the controls

plants (*Dex:GR*) sprayed with 1 μM of Dex, neither on *Dex:TT1* and *Dex:NTT* plants treated with low concentration of Dex (1 nM).

Cell death and senescence is characterized by cell membrane damage and electrolyte leakage[16]. To further test the effect of WIPs on cell death and senescence, leaves disks of *Dex:TT1*, *Dex:NTT*, and *Dex:GR* were collected after spray with 1 μM of Dex, and the changes of ion leakage were monitored at different intervals. The electrolyte leakage ($K^+$ efflux from plant cells) was measured in the suspension solution (water) at 1, 2, 4 and 24 h after Dex treatment. The obtained results indicate that overexpression of both *TT1* and *NTT* significantly increased ion leakage. However, the ion leakage in the *Dex:TT1* plants is much higher compared to *Dex:NTT* (Fig. 2e, f) and confirm the more severe leaf senescence phenotype observed in *Dex:TT1* plants. Taken together, these results suggest that TT1 and NTT are powerful transcription factors controlling plant growth and cell death.

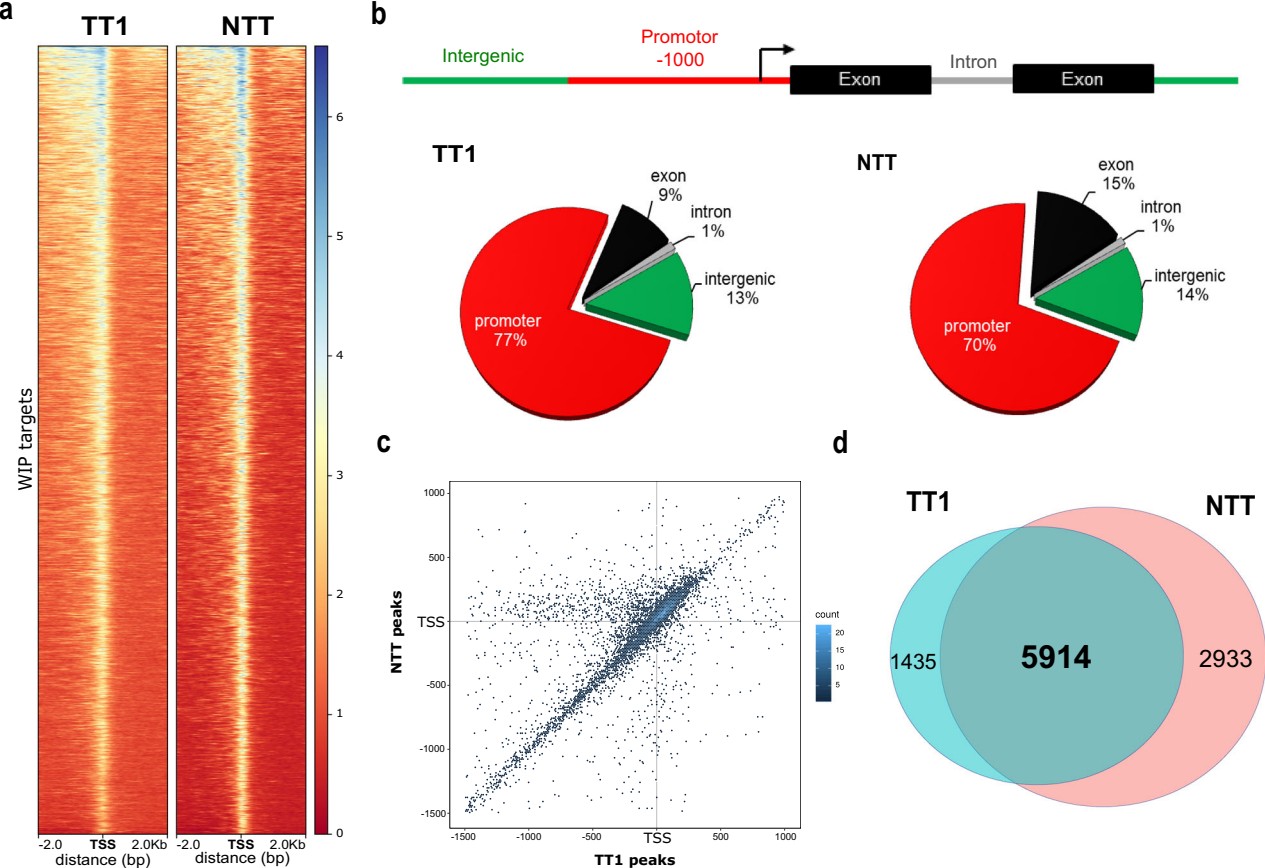

**Fig. 3 Genome-wide identification of WIPs binding sites. a** Comparison between *TT1-GFP* and *NTT-GFP* peak density in the region ±2 kb around the Transcription Start Site (TSS). **b** Representation of the distribution of *TT1-GFP* and *NTT-GFP* binding peaks identified by ChIP-seq in four different genomic regions (intergenic, promoter, exon, intron). Peaks between −1000 bp and the TSS were considered as located on the promoter region. Peaks located on 5′ UTR and 3′UTR regions were considered as part of the exon of the corresponding gene. The percentages of binding sites on each region are indicated. **c** Density plot showing overlap of *Dex:TT1* and *Dex:NTT* using hexagonal binning routine. Each point represents the distance of the nearest peak to the TSS of the corresponding gene. **d** The Venn diagram shows the overlap of *TT1-GFP* and *NTT-GFP* ChIP-seq target genes.

**Genome-wide identification of WIP binding sites**. To identify binding motifs and potential target genes of WIP transcription factors, we generated *Dex:TT1-GFP* or *Dex:NTT-GFP* transgenic plants and performed ChIP-seq analysis. Following Dex treatment, *Dex:TT1-GFP* and *Dex:NTT-GFP* transgenic lines accumulate *TT1* and *NTT* mRNA transcript as well as a strong GFP fluorescence signal (Supplementary Fig. 5c–e). Moreover, induced *Dex:TT1-GFP* and *Dex:NTT-GFP* lines show similar growth arrest phenotypes than lines without the GFP-tag (Fig. 2a) confirming that the GFP tag do not impair the TT1 or NTT function. Immunoprecipitation was done using an anti-GFP antibody on 7 days old seedlings treated with 1 μM Dex during 8 h. High-depth sequencing of the Illumina ChIP-seq libraries were generated and mapped (Supplementary Figs. 6a and 7). Peaks were distributed across the whole genome but strongly enriched in genic regions, especially in promoters, for both TT1 and NTT, comparing to input (Fig. 3a, b and Supplementary Fig. 8a). Most of the peaks were also overlapping and located around the transcription start sites (TSS) (Fig. 3c). Analysis of the ChIP-seq data led to the identification of 7349 and 8847 peaks associated with TT1 and NTT binding, respectively (Fig. 3d). To validate the ChIP-seq data we analyzed a common tagged gene of TT1 and NTT using ChIP-PCR, *WSIP2* (*WUS-INTERACTING PROTEIN 2*). Consistent with the ChIP-seq data, specific enrichment was obtained with primers at the ChIP-seq peak (Supplementary Fig. 8b, c). Comparative analysis of TT1 and NTT peaks led to the identification of 5914 genes with

overlapping binding sites (Fig. 3d and Supplementary Data 1). The strong overlap between TT1 and NTT target genes (80 and 67%, respectively) confirms the robustness of the ChIP-seq data and point out toward TT1 and NTT conserved functions. This is consistent with NTT and other WIPs complementing *tt1* mutant (Fig. 1c)[10]. This is also in line with the WIP overexpression lines sharing similar phenotypes (Fig. 2a) and the reported WIP functional redundancy (NTT, WIP4, and WIP5)[12].

A motif search analysis on the common binding sites of TT1 and NTT using Homer software[17] was then performed. Top three most significant binding motifs were identified as GAGA-repeat, bZIP (ABI5), and W-box (WRKY) motifs (Supplementary Fig. 8d). Interestingly a binding motif, targeted by a third WIP protein, AtWIP5, identified by DNA affinity purification sequencing (DAP-seq), was also listed among them on fifth position (Supplementary Fig. 8e)[18].

**Identification of the early target genes of WIP proteins**. To determine the gene regulatory networks recruited by *WIP* proteins to control growth, 7 days old seedlings of *Dex:TT1, Dex:NTT*, and the control line, *Dex:GR*, were treated with 1 μM Dex and used for gene expression analysis. High-depth sequencing of the Illumina RNA-seq libraries were generated and mapped. Normalized reads counts were analyzed using DEseq2 package (Supplementary Figs. 6b and 7). As expected, the expression of

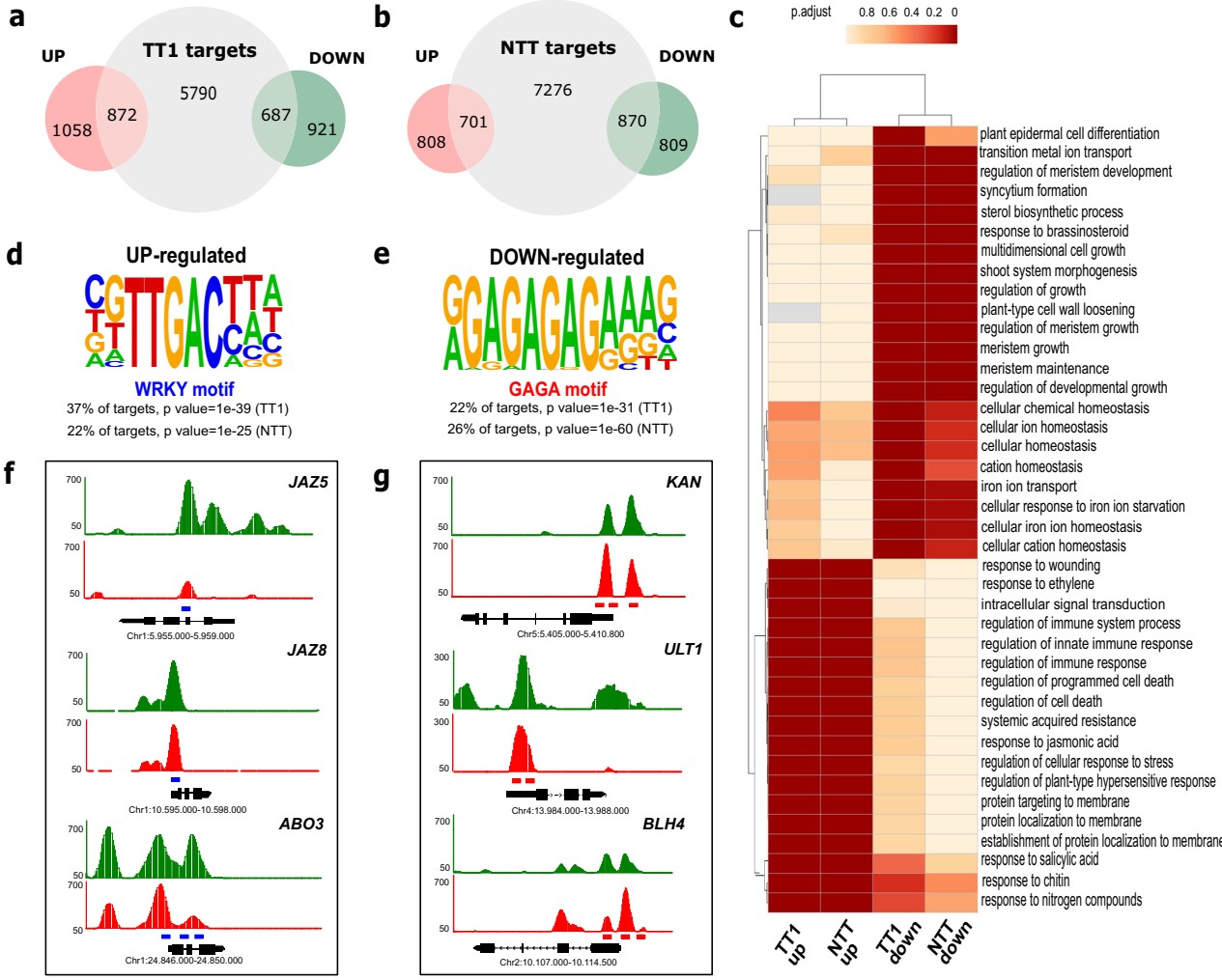

**Fig. 4 Integration of WIPs ChIP-seq and RNA-Seq data. a** The Venn diagram shows the overlap between targets and differential expressed genes identified after induction of TT1 with Dex. **b** The Venn diagram shows the overlap between targets and differential expressed genes identified after induction of NTT with Dex. **c** GO term enriched clustering analysis representing classification of genes directly targeted by TT1 and NTT depending on their expression profiles (up- or down-regulated). Color intensities represent significant values of each GO term. **d** Most highly enriched DNA-element identified in TT1 and NTT up-regulated target genes. **e** Most highly enriched DNA-element identified in TT1 and NTT down-regulated target genes. Representative binding profiles based on the IGB (Integrated Genome Browser) for *Dex:TT1-GFP* (green) and *Dex:NTT-GFP* (red) ChIP-seq data for selected genes whose expression was induced (**f**) or repressed (**g**) in response to Dex. A cartoon of *JAZ5, JAZ8, ABO3, KAN, ULT1,* and *BLH4* genes is shown below each panel, with boxes corresponding to exons and bars to intron. *Y*-axis corresponds to the height of the peak scaled to different values for each gene. The bars at the bottom of the peaks represent the motif locations determined by MACS2. WRKY and GAGA motifs are present on the promoter and exon regions of represented genes (blue and red boxes, respectively). Coordinates of each locus on the TAIR10 *Arabidopsis* Genome is indicated.

*TT1* and *NTT* was strongly induced after Dex treatment in the *Dex:TT1* and *Dex:NTT* lines, respectively, confirming the efficiency of the Dex treatment (Supplementary Data 2). Furthermore biological replicates formed distinct clusters from the control samples in the PCA-plot (Supplementary Fig. 9a).

Statistical RNA-Seq analysis identified 3538 and 3188 differentially expressed genes (DEGs) in *Dex:TT1* and *Dex:NTT*, respectively (FDR < 0.05, FC log2 ± 2.0). RT-qPCR confirmed the expression level of selected DEGs, showing that our RNA-seq data were reliable (Supplementary Fig. 9b). To identify co-regulated genes we compared the DEGs from both RNA-seq data. We found 1363 DEGs sharing similar differential expression profiles in *Dex:TT1* and *Dex:NTT* lines compared to *Dex:GR* control lines. Out of this, 761 genes were up-regulated and 602 genes were down-regulated on both lines. Interestingly, some genes (40 + 23) showed antagonistic expression in TT1 and NTT (Supplementary Fig. 9c and Supplementary Data 2).

In order to identify the set of primary WIP targets, we screened the DEGs for association with TT1 and NTT ChIP-Seq peaks. We observed significant overlap between those two different gene sets (Fig. 4a, b and Supplementary Data 3). Among the putative TT1-direct targeted genes, a slight higher number of genes were up-regulated (872, 24% of total DEG) comparing to those down-regulated (687, 19% of total DEGs); whereas on NTT, targeted genes were slightly more down-regulated (870, 27% of total DEGs) comparing to up-regulated genes (701, 21% of total DEGs). All together this data suggest that WIP proteins can act as a transcriptional activator or repressor.

To further investigate the main biological process in which WIP proteins are involved, we first performed Gene Ontology (GO) analysis, on shared DEGs that were also bound by TT1 and NTT. Consistent with WIP proteins sharing conserved function, most of the enriched GO terms, identified using clusterProfiler[19], were collectively common, in the TT1 and NTT-up and in the

TT1 and NTT-downregulated genes lists, respectively (Supplementary Data 4). GO terms related to meristem growth and maintenance, cell differentiation, ion homeostasis, and response to brassinosteroids were collectively overrepresented in the TT1 and NTT downregulated genes list (Fig. 4c). In contrast, GO terms related to pathogen resistance, including cell death, immunity, leaf senescence, and stress responses to hormones such as ethylene, JA, and salicylic acid (SA) were collectively overrepresented in the TT1 and NTT up-regulated genes list (Fig. 4c). We also performed GO analysis on TT1 and NTT-specific gene list. Consistent with the analysis of the shared gene list, all the downregulated genes were enriched in GO terms related to development and the upregulated genes were enriched in genes related to defense (Supplementary Fig. 10). For instance, GO terms related to seed dormancy process, plant ovule development, and adaxial/abaxial axis specification were significantly enriched in the TT1 down-regulated genes list (e.g., AGAMOUS-LIKE 14, AGL14; PETAL LOSS, PTL). On the other site, NTT down-regulated genes list was enriched in terms that relate to trichoblast differentiation, pigment accumulation and response to auxin (e.g., LEUCINE-RICH REPEAT/EXTENSIN 1, LRX1; AUXIN-REGULATED GENE INVOLVED IN ORGAN SIZE, ARGOS; PIN-FORMED 3 and PIN7).

Collectively, the expression profiles and GO analysis of TT1 and NTT targets point out on the role of WIP TFs in repressing development and growth by inducing plant defense response via hormonal variations.

## WRKY and GAGA motifs are predominant in WIP target genes.

Next we performed a motif search analysis on the binding sites of genes regulated by TT1 and NTT (combined ChIP-seq and RNA-seq analysis). Strikingly, we found that TT1 and NTT occupy overlapping binding sites, at probability far greater than what could have occurred by chance. The majority of the up-regulated targeted genes bound by TT1 (37%) and NTT (22%) were found to contain a TTGACY W-box (WRKY) motif with the statistical $p$-value of $1e^{-39}$ and $1e^{-25}$, respectively (Fig. 4d). W-box motifs are known targets of WRKY TFs, associated with stress and hormone signaling. As WRKY proteins, TT1 and NTT are also able to induce JA responsive genes such as JASMONATE-ZIM-DOMAIN (JAZ) PROTEIN 5 and JAZ8 by recognizing W-box motif in their promoters (Fig. 4f)[20]. WIP proteins also bind to the promoters of several WRKY genes such as ABA OVERLAY SENSITIVE MUTANT 3 (ABO3), which contain also W-box motifs that allow to autoregulate its own expressions, via a positive feedback loop[21].

In the other hand, the majority of the down-regulated targeted genes bound by TT1 (27%) and NTT (33%) contains GAGA-repeat motifs with the statistical $p$-value of $1e^{-31}$ and $1e^{-60}$, respectively (Fig. 4e). GAGA-motif binding factors (GAFs) could be divers, Trithorax-like and Pipsqueak protein families in animals, BASIC PENTACYSTEINE (BPC) protein family in plants, but all have a central function in growth and development[22–24]. It is also intriguing, that both animal and plant GAFs recruit members of the Polycomb Repressive Complex (PRC) to the Polycomb repressive DNA-elements (PREs) to silence the bound genes[23,25]. Similarly, genes involved on developmental process, such as KANADI (KAN), ULTRAPETALA 1 (ULT1), and BEL1-LIKE HOMEODOMAIN 4 (BLH4) were bound by TT1 and NTT, suggesting that WIPs act as transcriptional repressors through recognition of GAGA-motifs on their promoters (Fig. 4g).

In sum, integration of RNA-seq and ChIP-seq analysis reveals the interaction of WIP TFs with GAGA motif to repress the expression of growth and development promoting genes, and interaction with W-box motif to activate expression of genes associated with stress signaling pathways.

## Ectopic expression of TT1 in Arabidopsis carpel leads to male flowers.

To test the hypothesis that WIP proteins are general inhibitor of growth we ectopically expressed WIP proteins in two reproductive tissues, the carpel and the stamina primordia, and in two vegetative tissues, the trichome and lateral root primordia. Transgenic plants were generated and analyzed for organ growth inhibition.

We ectopically expressed TT1 in Arabidopsis carpel primordia using the tissue-specific CRABS CLAW (pCRC) promoter. CRC is a member of the YABBI family required for carpel development in Arabidopsis and its expression starts at stage 6 when the gynoecium primordium becomes distinct, disappearing at stage 10–11[26]. We obtained several independent pCRC:TT1 transgenic lines developing normal sepals, petals and stamens but with rudimentary arrested carpels (Fig. 5a, b and Supplementary Fig. 11a). Toluidine blue staining and scanning microscopy analysis showed normal development of carpel primordia at early developmental stages (stage 5), but from stage 8 carpel morphology was affected in pCRC:TT1 lines, given place to a rudimentary structure at later stages (stage 11), which finally leads to female-sterile flowers that mimic unisexual male flowers in melon (Fig. 5d, e and Supplementary Fig. 12a–h). Quantification of TT1 expression in young flowers of pCRC:TT1 transgenic lines correlated with the inhibition of carpel growth (Supplementary Fig. 11a).

To test whether ectopic expression of WIP gene can inhibit the development of other flower organs, we expressed TT1 under the control of APETALA3 promoter (pAP3). AP3 is a B class floral homeotic gene specifically expressed in the petal and stamens[27]. Again, in pAP3:TT1 lines a severe growth inhibition of stamen and petals was observed (Fig. 5a, c). Toluidine blue staining and scanning microscopy analysis showed that both petal and stamen development were completely inhibited in the pAP3:TT1 lines (Fig. 5d, f and Supplementary Fig. 11i–l). As expected, inhibition of growth correlated with the expression of TT1 in flowers of pAP3:TT1 transgenic lines (Supplementary Fig. 11b).

To test whether ectopic expression of WIP gene can also inhibit the development of vegetative tissues, we expressed TT1 under the control of the GLABRA-1 (pGL1) and the SOLITARY ROOTS (pSLR) promoters (Supplementary Fig. 11c, d). GL1 is a R2R3 MYB-type transcription factor expressed in young leaf primordia and developing trichomes[28]. Loss of function of GL1 leads to leaves without trichomes. Independent pGL1:TT1 transgenic lines were generated and all displayed serrated leaves with few or no trichomes (Fig. 5g–j). Inhibition of trichomes development also correlated with the expression of TT1 in leaves of pGL1:TT1 transgenic lines (Supplementary Fig. 11c). SOLITARY ROOTS gene encodes the IAA14 protein, a member of the Aux/IAA gene family that promotes lateral roots formation[29]. Loss of function of SLR leads to mutants that completely lack lateral roots and reduced root gravitropism. Three independent pSLR:TT1 transgenic lines were generated and none of them showed lateral root development after 4 weeks. As expected primary root elongation in the pSLR:TT1 lines was not affected (Fig. 5k, l). As for leaf phenotype and trichomes inhibition, lateral roots inhibition correlated with the expression of TT1 in lateral root primordia of pSLR:TT1 transgenic lines (Supplementary Fig. 11d).

In conclusion, our results demonstrate that TT1/AtWIP1 transcription factor could act as strong suppressor of organ development, as did CmWIP1 in carpel primordia of male melon flowers[6]. These data also suggest that WIP proteins could be recruited to inhibit growth of both vegetative and reproductive organs.

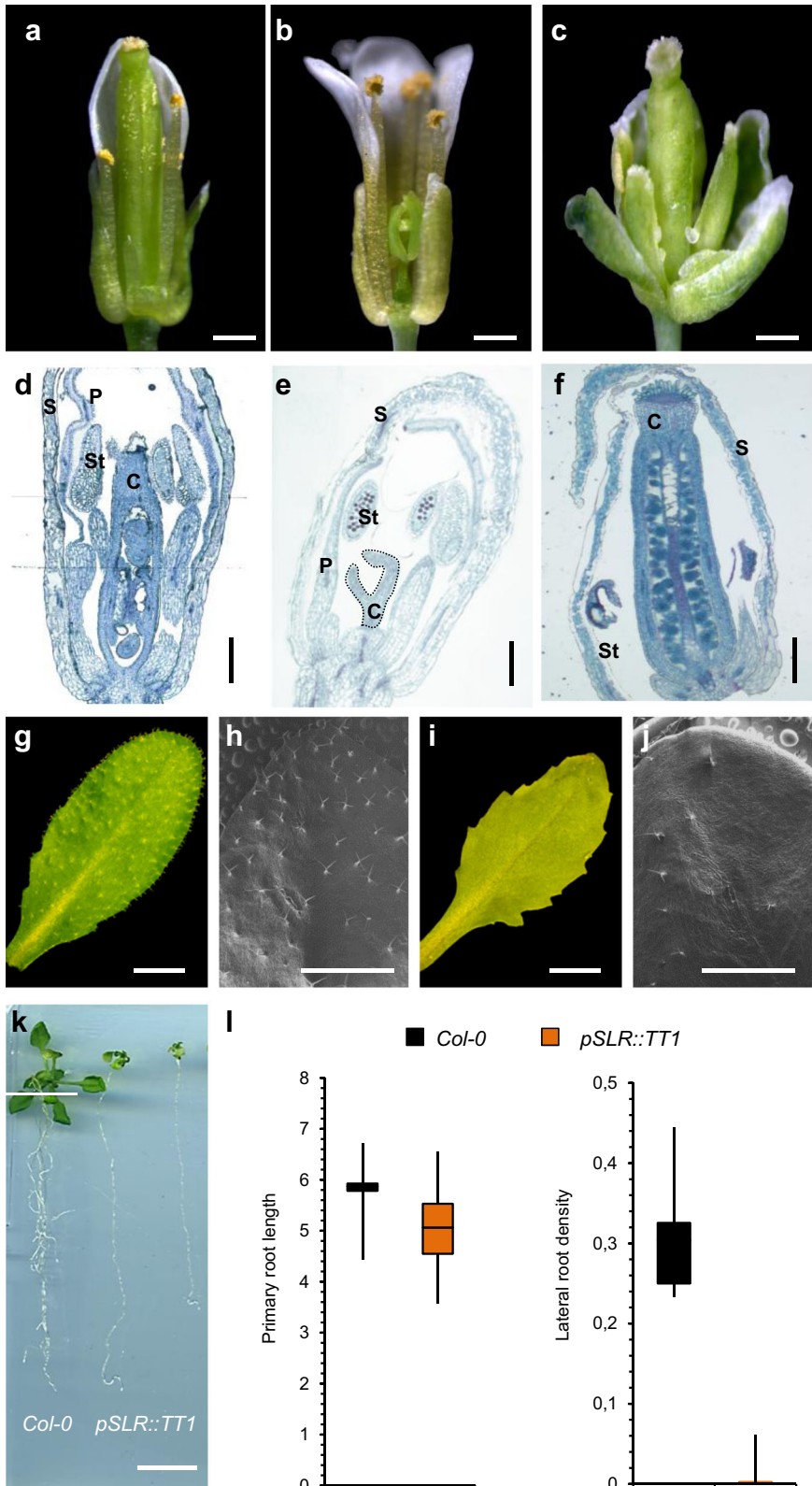

**Fig. 5 Ectopic expression of _TT1_ in reproductive and vegetative organs of _Arabidopsis_.** Phenotype of _Col-0_ (**a**), _pCRC:TT1_ (**b**), and _pAP3:TT1_ (**c**) transgenic flowers (scale = 0.5 mm). Toluidine blue staining of _Col-0_ (**d**), _pCRC:TT1_ (**e**), and _pAP3:TT1_ (**f**) flowers at late developmental stage (stage 10). Rudimentary carpel on _pCRC:TT1_ is designed with dotted line (s = sepals, p = petal, c = carpel, st = stamen; scale = 200 μm). Phenotype of _Col-0_ (**g**) and _pGL1:TT1_ (**i**) transgenic plants, which shows serrated leaves. Electronic microscopic analysis of leaves surfaces of _Col-0_ (**h**) and _pGL1:TT1_ (**j**) which shows reduced number of trichomes (scale = 2 mm). **k** Phenotype of _Col-0_ and _pSLR:TT1_ transgenic plants (scale = 1 mm). **l** Quantification of primary root length and lateral root density in _Col-0_ and _pSLR:TT1_ transgenic lines ($n = 8$, biologically independent seedlings).

## Discussion

WIP genes, $C_2H_2$ ZF transcription factors have different roles in land plants. In *Marchantia polymorpha*, MpWIP controls the development of the air pore complex[9]. In *Brassica napus*, BnTT1 is involved in the flavonoids and fatty acid biosynthesis[30]. In *Gerbera hybrida*, GhWIP2 controls petal and petiole's growth via regulation of cell expansion[31]. In *C. melo*, CmWIPl is a central integrator of expression networks leading to sex determination. Expression of *CmWIP* in carpel primordia inhibits carpel development, leading to unisexual male flowers[6]. In *Arabidopsis*, there are six WIP genes which show a strong amino acid conservation and were found to be functionally redundant[10]. WIP genes in *Arabidopsis* are involved in the accumulation of PA in the seed coat, development of replum and transmitting tract; and root meristem development[10–12].

Based on their sequence homology, we hypothesized that WIP genes might show conserved function across species and using a heterologous system might help to understand the molecular background of carpel arrest regulated by CmWIP1 in the non-model species, melon. We observed a significant restoration of the accumulation of PA in *tt1* mutants by expressing *CmWIP1*, suggesting that basic molecular functions are shared between these proteins (Fig. 1). Partial restoration was also obtained by Appelhagen et al.[10] in some cases (*prom*TT1:NTT and *prom*TT1:AtWIP3), however PA accumulation was less than 50% compared to WT seeds.

The transient and divers spatio-temporal expression patterns of WIP proteins during development have been a handicap for the characterization of the gene networks involving WIP proteins. Our results using an inducible promoter system, allowed us to synchronize the expression of two WIP proteins, NTT and TT1, in a particular tissue. Phenotyping of the Dex lines at seed germination and seedling stages demonstrated that WIP proteins are strong inhibitors of growth and triggers of cell death.

To identify gene regulatory networks recruited by *WIP* proteins we used RNA-seq and ChIP-seq analysis. By search for enriched GO terms in *TT1* and *NTT* data, we found several genes, associated with hormones stress signaling, up-regulated. This includes ethylene, ABA and JA signaling pathways, showing that hormone crosstalk mechanisms may be timely orchestrated by WIP proteins to limit defense-associated fitness costs. In contrast several genes involved in auxin transport (*PIN3*, *PIN4*, and *PIN7*) and gibberellins were downregulated by WIP. In a recent work, it has been shown that constitutive overexpression of *GhWIP2* (homologue of NTT in *G. hybrida*) caused major developmental defects associated with cell expansion including dwarfism, short petals, and petioles[31]. These phenotypes were also correlated with downregulation of gibberellins and auxin signaling genes and upregulation of abscisic acid signaling genes in *GhWIP2* overexpressing plants. All findings on different plant species converge into a conserved function of WIP proteins in integrating contrasting hormonal signals to regulate organ growth.

ChIP-seq analysis allowed us to identify two regulatory elements, W-box and GAGA motif, as binding sites of WIP transcription factors. Further integration of ChIP-seq, RNA-seq, and GO analysis revealed how WIP proteins could play a role as activators or repressors of transcription. We showed that WIP interact with GAGA motif to repress the expression of the growth and development promoting genes. It will be interesting to see whether WIP proteins also interact with of the Polycomb Repressive Complex to silence the bound genes, as described for GAFs in plants and animals[23,25]. W-box motif, known target of WRKY proteins that controls the expression of genes involved in defense-related phytohormones, such as SA or JA, imply a similar

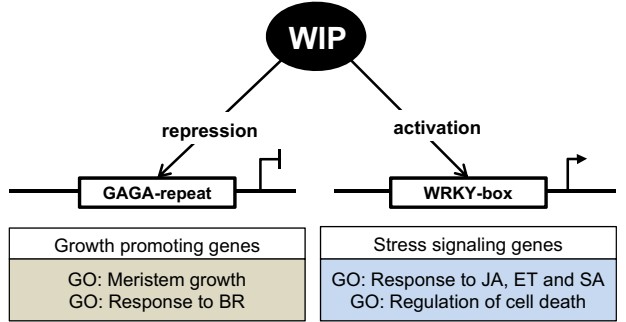

**Fig. 6 Schematic model of how WIP TFs control plant growth.** WIP TFs bound GAGA motif to repress the expression of the growth and development promoting genes, and W-box motif to activate expression of genes associated with stress signaling pathways.

role for WRKY and WIP genes in plant immunity (Fig. 6). From this we can also hypothesize that CmWIP1 in male flowers inhibits carpel development, through interaction with GAGA and W-box motifs. As melon is recalcitrant to genetic transformation it was difficult to tag CmWIP1 protein with an epitope that could be used for immunoprecipitation, required for ChIP-seq analysis. In silico screening of the promoter of genes differentially expressed between male and female flower, for the presence of GAGA and W-box motifs will be one way to identify CmWIP1 direct targets. Such targets could be further analyzed for differential expression in *gynoecious* plant compared to *androecious* plants and validated using genetic approaches.

In melon and related *Cucurbitaceae* species, *CmWIP1* and two ethylene biosynthesis limiting enzymes, *CmACS11* and *CmACS7* play a central role in sex determination. Expression of *CmACS7* requires carpel development and inhibits stamina development, leading to female flowers[5]. Expression of *CmWIP1* is under the negative control of ethylene produced by *CmASC11* in flowers developing in young vines. The non-expression of *CmACS11* permits the expression of *CmWIP1* which inhibits the carpel development, leading to male flowers. To test whether expression of WIP proteins can also lead to male flowers in *Arabidopsis*, we disconnected the control of the expression of the WIP protein of the ethylene signaling, using *CRC* carpel specific promoter. In melon, sex determination occurs at stage 4–6 of flower development[4]. The choice of *CRC* to drive the expression of WIP protein in *Arabidopsis* is justified by its early expression during carpel development, stage 6, and because it is very specific for nectary and carpel development. When *TT1* was ectopically expressed under *CRC* promoter in carpel primordia in *Arabidopsis*, it led to male flower formation, mimicking *CmWIP1* function in melon.

Interestingly, ectopic expression of *TT1* in stamina primordia, as well as in trichomes and lateral roots, demonstrated that TT1 inhibition of growth is not only confined to the carpel, but was also to other reproductive and vegetative organs. In contrast, it was previously shown that ectopic expression of *NTT* under the control of the *ASYMETRIC LEAVES 1* (*AS1*) promoter promote formation of roots from cotyledons primordia instead of causing growth arrest as we observed with *TT1*[12]. Whether, TT1 and NTT have different functions or the level of expression obtained by the chosen promoters (extremely low or extremely high) is different between those experiments, we may consider WIP genes as regulators of cell fate and differentiation. Indeed, in our RNA-seq data, we have identified a set of genes that are significantly different with only one of the AtWIP, suggesting that those genes have contributed to acquire specific roles of TT1 and NTT during

evolution. For instance, several genes involved in cytokinin pathway showed opposite behavior after *TT1* and *NTT* induction.

Our results have highlighted the evolutionary biology of WIP genes. Throughout the species, WIPs genes tend to shape development through several pathways. As we have observed, following expression of WIP proteins, up-regulation of stress inducible genes and down-regulation of growth and development associated genes, this study has initiated another debate over the complex function of WIPs. This study has also highlighted the role of WIP proteins in crosstalk between hormones associated with stress and the one associated with growth. The difference of genetic regulation of the WIP transcription factors emphasizes on the difference of their particular role in the plant development. A substantial work is still required to complete the puzzle of functional characterization of the WIP transcription factors in the plant kingdom. Still because we were able to engineer unisexual flowers in *Arabidopsis*, using discoveries in melon, our work highlight how the WIP proteins could be used as biotechnology tool to control crossing and production of F1 hybrids seeds in general.

## Methods

**Plant material**. *Arabidopsis* (*A. thaliana*) genotypes (accession *Columbia-0, Col-0*) were grown under a 16-h-light/8-h-dark condition at 22 °C in a growth room. For seed germination, sterilized seeds were incubated at 4 °C for 2 days and sown on Murashige and Skoog (MS) medium plates containing 1% sucrose and 0.8% agar. *tt1-3* mutants (SALK_026171) (*Col-0* background) were previously characterized[10].

**Plasmid construction and *Arabidopsis* transformation**. To construct the binary vector *35S:CmWIP1, CmWIP1* cDNA from *C. melo* was amplified with iProof® DNA polymerase (Bio-Rad) and cloned into *pDONR-207* and then inserted in the binary vector *pGWB2* using the Gateway cloning system with LR clonase II and transformed in *Arabidopsis* using *Agrobacterium tumefaciens* strain C58C1. Primers used are listed in Supplementary Data 5.

For both, the promoter swap and the ectopic expression experiments, the protocol used was the following: 2 kb of *TT1* promoter, 727 bp of *AP3* promoter, 3.8 kb of *CRC* promoter, 2 kb of *SLR* promoter, and 1.5 kb of *GL1* promoters were amplified using genomic DNA as template. For all of them, PCR was done with a reverse primer having SpeI restriction site. After digestion with SpeI (New England, Biolabs) the promoters were cloned into *pIJPB* binary vector[32] and confirmed by sequencing. The *AtTT1* CDS was amplified with iProof® DNA polymerase (Bio-Rad) from cDNA and cloned into *pDONR-207*. *AtTT1* or *CmWIP1* CDS were then introduced into the pIJPB plasmid by Gateway® system (Invitrogen™). Transformation of *Arabidopsis* was done using *A. tumefaciens* strain C58C1 by floral dip method. Primers used for cloning and validation of these lines are listed in Supplementary Data 5.

*Dex:TT1* and *Dex:NTT* constructs were produced as follows: *TT1* and *NTT* CDS were cloned into the donor vector (*pDONR221*) (Life Technology). *TT1-GFP* and *NTT-GFP* sequences were generated by chimeric PCR and then cloned into the donor vector. *TT1, NTT, TT1-GFP*, and *NTT-GFP* were then introduced into the Gateway destination vector *pEN-L4-pOp6M2-R1* which contain the *pOp6* promoter that carry six copies of a lac operator sequence[33]. *pOp6:TT1, pOp6:TT1-GFP, pOp6:NTT*, and *pOp6:TT1-GFP* constructs were transformed in *Arabidopsis* seedlings carrying the *35S::LhG4* vector and T1 generation seeds were selected for Basta and Kanamycin resistance. Primers used for cloning and validation of these lines are listed in Data S5.

**Phylogenetic analysis**. Phylogenetic analysis of WIP proteins from *A. thaliana* and *C. melo* was performed using the software MEGA (Molecular Evolutionary Genetic Analysis; version 7.0.20)[34]. Sequences were aligned with ClustalW method, and phylogenetic tree was obtained by neighbor-joining (NJ).

**Conductivity assays**. Senescence and associated cell death were estimated as previously described[35], with minor modifications. Briefly, ion leakage was measured from 5-week-old plants of *Dex:TT1, Dex:NTT*, and control (*Dex:GR*) sprayed with different concentrations of Dex (0.001, 0.05, 0.1, and 1 μM). Three leave disks were collected after 24 h treatment and placed in 2 mL sterile water at room temperature and conductivity (Ci) was measured at 0, 1, 2, 4, and 24 h using an ion conductivity meter (B-173, Horiba, Kyoto, Japan). To calculate final electrolyte conductivity (Ct), leave disks were then placed at −20 °C for 1 h, and at 70 °C for 3 h to induce 100% ion leakage. Relative ion leakage was calculated for each time point and expressed as a percentage of the total ion leakage (% = Ci/Ct × 100).

**Vanillin staining and proanthocyanidins quantification**. Immature seeds (heart stage of embryogenesis) of *Arabidopsis* were stained with 1% vanillin (4-Hydroxy-3-methoxybenzaldehyde) (Sigma-Aldrich) in 6 N Hydrochloric acid for 30 min and observed with Leica macroscope. PA quantification was carried out according to Appelhagen et al.[10]. Approximately 10 mg of mature seeds were grinded with 200 μL of acidic methanol. 1200 μL acidic BuOH was added followed by 40 μL "ferric reagent" mix. Tubes were boiled at 95 °C for 20 min and then cooled for 5 min on ice. Spinning was done at 13,000 rpm for 4 min. Supernatant was used to measure the UV absorption (550 nm) and compared to Cyanidin Chloride standard (Sigma-Aldrich) curve for quantification.

**ChIP-seq and ChIP-PCR analysis**. ChIP assays were performed as described by Jégu et al.[36]. Briefly, 7 days old seedlings of *Dex:TT1-GFP* and *Dex:NTT-GFP* lines were transferred to MS liquid medium supplied with 1 μM Dex. After 8 h treatment, seedlings were crosslinked with 1% formaldehyde and the reaction was stopped with glycine (2 mM). Chromatin was sonicated to an average length of 300 bp and immunoprecipitated (overnight at 4 °C) with an anti-GFP antibody (Clontech 632592). The chromatin was then reverse-crosslinked (RNase and proteinase K digestion) and DNA was extracted with phenol-chloroform. Precipitated DNA (10 ng of IP or input DNA) was used for ChIP-seq library construction using a NEBNext® Ultra DNA Library Prep Kit for Illumina®. Single-end sequencing of IP and input samples were performed using GAIIx Illumina® (read length of 50 bp). In total six libraries were sequenced, one for *Dex:TT1-GFP* and one for its input and two for *Dex:NTT-GFP* samples and two for their corresponding inputs. Reads were mapped with BWA (Burrows–Wheeler Aligner) onto *Arabidopsis* genome release 43 from Ensembl database[37], then only reads with mapQ > 30 and uniquely mapped were kept using Samtools (http://www.htslib.org/). Reads identified as PCR duplicates were discarded using Picard Toolkit (https://broadinstitute.github.io/picard/).

The MACS2 (version 2.1.2) software was used to identify significantly enriched regions ($q$-value < 10$e^{-10}$)[38]. Visualization and analysis of genome-wide enrichment profiles were done with IGB (Integrated Genome Browser, https://bioviz.org/). Peak annotations were assigned using HOMER, keeping only peaks in an interval of [−1500; +150] around the TSS (Transcription Start Site). Heatmaps were drawn using deeptools software[39]; hexbin plot with R[40] and ggplot2 library[41]. Putative binding peaks obtained from ChIP-seq analysis were confirmed by ChIP-qPCR assays using anti-GFP (Clontech 632592) and anti-IgG antibody (Millipore) antibodies. The primers used for ChIP-qPCR assays are listed in Supplementary Data 5.

**RNA-Seq analysis**. Seven days old seedlings of *Dex:TT1, Dex:NTT*, and *Dex:GR* lines were treated with Dex (1 μM during 8 h), then grinded and total RNA was extracted with RNAeasy plant kit and treated with DNaseI (Qiagen, Germany). Eight samples were used for library preparation (two biological replicates per genotype), comprising *Dex:TT1* and its corresponding control (*Dex:GR*) lines, and *Dex:NTT* and its corresponding control (*Dex:GR*) lines. Multiplex Illumina sequencing using DNA barcoding libraries were prepared using the TruSeq RNA Library Preparation Kit Illumina® according to the manufacturer's recommendations. Libraries were sequenced on the HiSeq2000 platform Illumina®, and between 30 and 50 million read pairs per sample were obtained. Quality was assessed using FastQC (version 0.11). Tophat2 (version 2.1.0) has been used to generate the mapping files and RSeQC aligned reads above 94% of read pairs of each sample correctly to TAIR (version 10). The mapped reads were assigned to genes with featureCount (v1.5.0-p3). DESeq2 (version 1.10.1) analysis was employed for differentially expressed gene calling ($p$-value adjusted > 0.05 and log2 FC < −2 and >2).

**GO analysis, motif search, and gene network construction**. Motifs search was performed by HOMER with TAIR10 motif database[17]. GO enrichment analysis was carried out with Rstudio (http://www.rstudio.com/) and ClusterProfiler package[19]. Euler diagram were computed with R and eulerr package[42]. Heatmap was drawn with R package Pheatmap (https://rdrr.io/cran/pheatmap/).

**Microscopy**. Plant tissues were fixed in the fixation buffer (2% paraformaldehyde, 1% glutaraldehyde, 1% caffeine, 0.1 M phosphate buffer, pH 7). Samples were dehydrated with 70, 95, and 100% ethanol and embedded in resin Technovit® 7100 (Kulzer). 10 μm sections were cut with a microtome Leica RM2165 equipped with cutting glass, mounted on the glass slide and stained with 0.05% (w/v) toluidine blue. Fresh leaf samples and inflorescence were observed with Tabletop SEM SH-1500MB electron microscope. *Arabidopsis* roots were stained with 0.1 mM propidium iodide for 15 min as previously described[43] and analyzed in a Leica TCS-SP2-AOBS spectral confocal laser scanning microscope (Leica Microsystems).

**Expression analysis**. For RT-qPCR, total RNA was extracted with RNeasy plant kit and treated with DNase I according to the manufacturer's instructions (Qiagen, Germany). Reverse transcription was done with SuperScript II RT (Invitrogen, Germany). Quantitative PCR was done with Bio-Rad CFX96 Real-time PCR system using SYBR® Green no ROX (Eurogentec). Relative expression for all genes was normalized to AtACT2 and as internal reference. Relative expression was

calculated with the relative −ΔΔCt method. Primers used are listed in Supplementary Data 5.

**Statistics and reproducibility**. For phenotypic analysis, at least two independent transgenic lines for each construct were analyzed. For PA quantification, six measurements for each line were realized using 10 mg of mature seeds. For ion leakage measurements two independent experiments were performed. For each experiment, four plants per genotype and per treatment were recorded. Per time point a pool of 3 leave disks were collected per plant. *T*-test was calculated using the two-tailed test function in excel.

For RNA-seq and ChIP-seq, pools of 7 days old seedlings have been used as biological replicate. Pools contained around 100 seedlings grown on different Petri dishes randomly distributed in the growth chamber and treated independently with dexamethasone. Two biological replicates were performed for RNA-seq. DEseq2 algorithm was used to define the significant transcripts for each genotype (*Dex:TT1* and *Dex:NTT*) against its corresponding controls. Three ChIP-seq experiments were performed, two biological replicates for *Dex:NTT* and one for *Dex:TT1*. Only peaks that were discovered above the background (corresponding to input) were considered for further analysis.

For RT-qPCR analysis, the same RNAs used for RNA-seq were analyzed. The expression values correspond to the average of two biological replicates, with three technical replicates. For ChIP-PCR, immunoprecipitated chromatin was prepared from new samples, and the average corresponds to three technical replicates.

**Reporting summary**. Further information on research design is available in the Nature Research Reporting Summary linked to this article.

## Data availability

The raw RNA-seq and ChIP-seq data described in this study have been deposited to the NCBI Short Read Archive (SRA) database under the BioProject ID PRJNA608903. Sequence data from *CmWIP1* can be found in the GenBank data library under accession number GQ870274.1. Sequence data from *TT1/AtWIP1* (AT1G34790), *NTT/AtWIP2* (AT3G57670), *AtSTOP1* (AT1G34370), *CRC* (AT1G69180), *AP3* (AT3G54340), *GL1* (AT3G27920), and *SLR* (AT4G14550) can be found at The Arabidopsis Information Resource (TAIR) database.

## Code availability

The code and software sources from previously published papers are listed in the "Methods". All other details of the code used in this study are available from the corresponding author upon request.

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

## Acknowledgements

This work was supported by the European Research Council (ERC-SEXYPARTH, 341076), the ANR (EPISEX, ANR-17-CE20-0019), and the LabEx Saclay Plant Sciences-SPS (ANR-10-LABX-40-SPS). M.V.G.R. was supported by the Intra-European Fellow-ships for Career Development (IEF) (Grant PIEF-GA-2012-330908). The authors thank Ikram Blilou for fruitful discussions, the research facilities provided by the Institute of Plant-Science Paris-Saclay (IPS2, France) and the Pôle Scientifique de Modélisation Numérique of the ENS (Lyon, France) for computing resources.

## Author contributions

M.V.G.R., A. Boualem, and A. Bendahmane designed the experiments and supervised the study. M.V.G.R., F.I., P.C., A.H., V.S., J.E., D.L., T.J., M.V., and J.S. performed and/or analyzed the data. M. Benhamed and M. Bendahmane helped with the interpretation of the data. M.V.G.R., A. Boualem, and A. Bendahmane wrote the manuscript. All authors revised the draft manuscript and read and approved the final manuscript.

## Competing interests

The authors declare no competing interests.
