## [Peer review file · Communications Biology]

Reviewers' comments:

Reviewer #1 (Remarks to the Author):

The review of the manuscript:

“ Integrative genome-wide analysis reveals the role of WIP proteins in inhibition of growth and development”

The Authors investigated WIP proteins, in the Arabidopsis model. We use a series of indirect transformation methods to obtain a number of plants that they have studied in detail. Authors showed that WIPs are an inhibitor of growth and inducer of cell death. Additionally using ChIP-seq and RNA-seq they revealed that most of the up-regulated genes bound by WIPs display a W-box motif, associated with stress signaling and the down-regulated genes contain a GAGA motif, a known target of polycomb repressive complex. It was also demonstrated that WIPs can trigger the growth arrest of vegetative and reproductive organs.

The aim of the study was to clarify the role of WIP proteins in Arabidopsis and an aspect of sex determination in melon. The authors conducted extensive analyses of the studied genes, The strong points of the manuscript are a very interesting topics about the function of WIP proteins, functional transformation and application of NGS technology.

Noteworthy are the multitude of results and nice documentation through Figures. However, I lack a good description of materials and methods.

The current guidelines for expressional analysis are that experiments using RNA - Seq data to describe changes should contain the biological replicates (unless otherwise justified) herein authors do not indicate this in the material and methods section. Each biological replicate should be represented in an independent library, each with unique barcodes if libraries are multiplexed for sequencing. In this case, the authors did not specify if the libraries were prepared independently. Actually, the information given by the author in the manuscript about the replicas is not clear. It is good practice to confirm the RNA-seq analyzes by qPCR, but the authors did not provide data on this topic.

When performing transformation experiments, it is worth checking the presence of the transgene in DNA as well as its expression. The authors do not submit results for these confirmations.

Additionally, the full data accompanying these experiments must be made available to reviewers at the time of submission in a freely accessible resource e.g. SRA database or others. The authors did not provide any information about the submission.

At the end of the general comments, because of the complex experiments, I suggest you add a flow diagram to make the work easier to read.

Line 90 – how many family members are in Melon? It is already known or not? The genome of melon is already sequenced and WIP genes were also investigated in this species. This occurs later, but also should be mention in the introduction

Line 97 - 110- In this paragraph, it is worth mentioning these six WIP transcription factor family members and state what is their common denominator. Is the construction about which the authors write above line 87-89 or something else? This paragraph needs some rebuilding.

Line 112 - 121- It is not clear why the authors chose these two genes for analysis of the six present in Arabidopsis. Could you give the reason for this choice here?

Line 360 – 363 - This conclusion is not justified in the results obtained. What is carried out on Arabidopsis does not necessarily translate into such a direct way to another species in this case – melon? Besides, the authors themselves state in the first words of the discussion (Line 366 – 369) that WIP genes play different roles in different plants.

Line 435 – 439 First of all, you should specify which plant you work with. Can you indicate in this paragraph that it is Arabidopsis?

Line 440 – The title of this: “Arabidopsis Plasmid construction and transformation” better “Plasmid construction and Arabidopsis transformation”.

Line 456 - “a the” too much of something

Line 488 – 500 Have any biological or technical replicates been used in the CHIP-seq and CHIP-PCR analyzes?

Line 502 – Please provide information about IGB

Line 504 - Please provide information about TSS

Line 509 – 519 Have any biological or technical replicates been used in the RNA-seq analyzes?

Line 535 – 537 Have any biological or technical replicates been used in the Quantitative PCR analyzes? What about statistics in this case?

Line 538 - Is the sequential data obtained by you available e.g. in the SRA or other databases?

Figure 6 - Do these interactions mean something? Is this just a simple graphic? This is unclear and misleading.

Reviewer #2 (Remarks to the Author):

This manuscript wants to clarify the molecular background of carpel arrest regulated by WIP1 gene and hypothesizes that WIP genes show conserved function across species. Large volume of transgenic work which benefit to understand the gene function was contained in this work. With these OE lines, the authors analysis the genome-wide WIP binding site and clarify the related pathways. The results show that WIP TFs repress development and growth by inducing plant defense response via hormonal variations.

In general, the manuscript reports interesting findings. The text is easy to follow and most of the figures are clear. Here, I have some questions:

1.As we all know, the ethylene pathway together with the WIP1 gene decide the sex expression in Cucurbitaceae, but the detailed mechanism remains to be elucidated. Based on your data, could you provide some explanations? Or the mechanism is different in Arabidopsis?

2.The gene expression patterns of those WIP members should be added.

3.In melon and watermelon, the WIP1 is an apical shoot-specific gene, and the expression of WIP1 is detected in the flower buds at stages 1 to 5 before they diverge into the male or female organ. As shown in your work, the CRC promoter was expressed at stage 6 in carpel. Maybe you can add another earlier expressed promoter to compare the phenotype.

4.In your schematic model,WIP has a direct interaction with DNA, but there is no data to support. And how to explanation both WRKY and GAGA motifs can bind to WIP?

Minor points about the figures :

1.In Fig.1A,why and how to divide the phylogenetic tree into two part (pink and blue background)?

2.In Fig.1D, two transgenic lines (#6, #11) were shown. But in B and C , only one line, and which one?

3.3. In Fig 3, it is hard to understand that the target gene of WIP is more than 6000. Any explanation?

Reviewer #3 (Remarks to the Author):

In this report, Roldan et al. unveiled a hidden commonality in the function of WIP genes, using Arabidopsis as a model. The authors started from the mystery that CmWIP1, which can act for sex determination in melon, is nested into the gene categories with the functions relating seed color or fertilization of silique in Arabidopsis. They showed that the fundamental protein functions of these WIP family genes are almost identical, where overexpression can act for repression of the growth and senescence of leaves, with a few exceptions. Combination of ChIP-Seq and transcriptomic data suggested the same molecular function of these WIPs to recognize the common cis-motif and to regulate similar pathways. Lastly, the authors showed the possibility that these WIPs probably can work for organ-specific repression depending on the promoters, which might explain the sex determining function in Cucumis. The methods sounded, and the conclusion was convincing and very clear. I really enjoyed reading this manuscript, although I would raise some suggestions and questions on the contents of this manuscript, as followings.

1. CmWIP1 would have a very important position to explain plant sexuality based on ethylene signaling. Thus, if the authors avoid the discussion about CmWIP1, the manuscript might be only one of the many papers involving Arabidopsis transcription factors. However, although the authors started from CmWIP1 in melon, the discussion (not the results) was focused only in Arabidopsis situations. I strongly recommend to write about the (estimated) regulatory systems of CmWIP1 via ethylene signaling (or ACS regulation) to determine sexuality in melon. I guess that the part on the WRKY motif (Figure 4) and the last part (Figure 5) might be directly related to the discussion of sex determination in melon. CmWIP1 can recognize W-box in ACS-7 in androecium/gynoecium specific manners?

2. TT1-induced Arabidopsis under the control of CRC- or AP3-promoters showed clear conversion into male and female flowers. Partially related to the question above, were their regulatory networks similar to melon's male and female flowers? If the authors concept that WIP genes potentially carry identical trans-functions (and the function differentiation may be dependent on cis-regulation), these sex-converted Arabidopsis would be able to reflect the expression differentiation between male/female melon.

Minor points

1. Figure 1A: Please add outgroup gene and genes from other species, for definition of the distinct evolutionary clades. The current clustering looks very arbitrary.
2. In some pictures, such as Figure 2C-D, 5A-C, and 5G-K, scale bars missed.
3. Figure 4F-G: quite confusing figures. Please add X- and Y-axis scales. For my viewer, bars in the peaks are missing. The authors should add more interpretations of these panels.

The review of the manuscript:

“ Integrative genome-wide analysis reveals the role of WIP proteins in inhibition of growth and development”

Reviewers' comments:

Reviewer #1 (Remarks to the Author):

Comment 1: The Authors investigated WIP proteins, in the Arabidopsis model. We use a series of indirect transformation methods to obtain a number of plants that they have studied in detail. Authors showed that WIPs are an inhibitor of growth and inducer of cell death. Additionally using ChIP-seq and RNA-seq they revealed that most of the up-regulated genes bound by WIPs display a W-box motif, associated with stress signaling and the down-regulated genes contain a GAGA motif, a known target of polycomb repressive complex. It was also demonstrated that WIPs can trigger the growth arrest of vegetative and reproductive organs.

The aim of the study was to clarify the role of WIP proteins in Arabidopsis and an aspect of sex determination in melon. The authors conducted extensive analyses of the studied genes, The strong points of the manuscript are a very interesting topics about the function of WIP proteins, functional transformation and application of NGS technology. Noteworthy are the multitude of results and nice documentation through Figures.

In the name of the authors I would like to thank Reviewer 1 for acknowledging the importance of this work.

Comment 2: However, I lack a good description of materials and methods. The current guidelines for expressional analysis are that experiments using RNA - Seq data to describe changes should contain the biological replicates (unless otherwise justified) herein authors do not indicate this in the material and methods section. Each biological replicate should be represented in an independent library, each with unique barcodes if libraries are multiplexed for sequencing. In this case, the authors did not specify if the libraries were prepared independently. Actually, the information given by the author in the manuscript about the replicas is not clear.

As requested by the reviewer and for more clarity we have included more detail in the methods section. For instance libraries for RNA-seq were prepared and sequenced at all together using specific barcodes for each library. In total 8 libraries were sequenced, 2 x *Dex:TT1* and 2 x *Dex:GR* (control for TT1); and 2x *Dex:NTT* and 2x *Dex:GR* (control for NTT). CHIP-seq libraries for *Dex:TT1-GFP* and *Dex:NTT-GFP* were prepared independently. In total six libraries were sequenced using multiplexing, one for *Dex:TT1-GFP* and one for its input and two for *Dex:NTT-GFP* samples and two for its corresponding inputs. In addition we added a new figure, (Figure S7) that describes the full workflow of ChIP-seq and RNA-seq samples preparation and data analysis. Additional information on the libraries and the number of reads are describes in Figure S6.

Comment 3: It is good practice to confirm the RNA-seq analyzes by qPCR, but the authors did not provide data on this topic.

In our submitted version we have validated our ChIP-seq and RNA-seq data by qPCR analysis. For more clarity we have included now, in supplementary data, the qPCRs analysis of 9 selected genes that validate further the RNA-seq data (see Figure 9B). Validated ChIP-seq is indicated in Figure S8 B and C.

Comment 4: When performing transformation experiments, it is worth checking the presence of the transgene in DNA as well as its expression. The authors do not submit results for these confirmations.

Because of the large number of transgenic plants produced we have put only the information that we believed important for the readers. For instance for *Dex:TT1*, *Dex:NTT* and *Dex:GR*, Figure 2B show qPCR data quantifying the overexpression of TT1 and NTT after Dex induction.

As requested by the reviewer we have now added details on all the generated transgenic plants. For all the transgenic lines only the lines showing single insertion segregating in Mendalian fashion were analyzed.

For instance, in Figure S3, we have add three panels (see new figure S3A-C) that describe the *35S:CmWIP1* insertion site in Arabidopsis genome, determined by the sequencing of the flanking sequence tag (FST), and PCR results showing the T-DNA insertion and expression of the transgene in the transgenic lines.

In Figure S4 we have added genotyping and expression data of the Arabidopsis *ttl* mutant-transgenic plants expressing *CmWIP1* coding sequence under the control of *TTI* promoter. The presence of the T-DNA and the expression of the transgene in independent transgenic lines is shown.

In Figure S5, we add two panels that describe the analysis of independent *Dex:TT1-GFP* and *Dex:NTT-GFP* transgenic lines for presence of the transgenes. PCR fragments validating the presence of LhGR, pOp6:TT1 and pOp6:NTT or the WIP fused to GFP T-DNAs are shown. Idem for *Dex:TT1* and *Dex:NTT* (see Figure S5 B)

For ectopic expression (*pCRC:TT1*, *pAP3:TT1*, *pGL2:TT1* and *pSCR:TT1*), we have added PCR results showing the T-DNA insertion and quantification of the expression of *TTI* compared to WT plants (see Figure S11A-D).

Comment 5: Additionally, the full data accompanying these experiments must be made available to reviewers at the time of submission in a freely accessible resource e.g. SRA database or others. The authors did not provide any information about the submission.

We have deposited all the data in public database and planned to add the codes in the accepted version of the paper; we apology for not putting them at the start of the submission. We are providing the code numbers in the resubmitted version.

Comment 6: At the end of the general comments, because of the complex experiments, I suggest you add a flow diagram to make the work easier to read.

We would like to thank the reviewer for this suggestion. In the new submitted version, we include a figure that describes the workflow of our experiments (see Figure S7). This clearly will help specialized readers that would like to deeply analyze again our omics data.

Comment 7: Line 90 – how many family members are in Melon? It is already known or not? The genome of melon is already sequenced and WIP genes were also investigated in this species. This occurs later, but also should be mentioned in the introduction. Line 97 - 110- In this paragraph, it is worth mentioning these six WIP transcription factor family members and state what is their common denominator. Is the construction about which the authors write above line 87-89 or something else? This paragraph needs some rebuilding.

The recent draft of the melon genome counts six melon WIP genes including *CmWIP1* (which is wrongly annotated on two pseudogenes). Blasts analysis of WIP genes in melon or other angiosperms have been never been published and our phylogenetic analysis demonstrated the sequence correspondences with *Arabidopsis* WIPs (see Figure 1A). As requested by the Reviewer, we have included a short sentence in the introduction to address this point.

Concerning the common denominator of the WIP proteins, the following paragraph in the introduction clearly explain what we know this far on this sub class of zinc finger transcription factors:

<<CmWIP1 belong to subclass A1d of zinc finger transcription factors, characterized by a highly conserved C-terminal domain and two conserved motifs in the N-terminal^{7,8}. In the C-terminal, there are three conserved amino acids, tryptophan (W), isoleucine (I) and proline (P) on which the protein family is named WIP, followed by four C₂H₂ zinc finger (ZF) motifs. WIP proteins are land plants specific transcription factor, present in one copy in the liverwort *Marchantia*⁹, two copies in the moss *Physcomitrella*, three copies in the genome of the lycophyte *Selaginella moellendorffii* and six copies in most diploids (such as *Arabidopsis*, tomato and melon) and higher copies in polyploids angiosperm. So far, little is known about the molecular function for WIP proteins and no transcriptomics analysis have been done that allows identification of the main gene targets of this family of transcription factors.>>

Comment 8: Line 112 - 121- It is not clear why the authors chose these two genes for analysis of the six present in *Arabidopsis*. Could you give the reason for this choice here?

Our phylogenetic analysis point to two separate clades of WIP proteins. Clade one is made of NTT, WIP4 and WIP5. The second clade is made of TT1, WIP3 and WIP6. We have selected TT1, because it is representative of clade 2 and the phenotype of the mutant is easily to score (*transparent testa*). We also selected NTT because it is the nearest homologue to the melon sex gene from the second clade (see figure 1 panel A)

Comment 9: Line 360 – 363 - This conclusion is not justified in the results obtained. What is carried out on *Arabidopsis* does not necessarily translate into such a direct way to another species in this case – melon? Besides, the authors themselves state in the first words of the discussion (Line 366 – 369) that WIP genes play different roles in different plants.

We have stated in our original submitted version: << In conclusion, our results demonstrate that AtTT1/WIP1 transcription factor act as strong suppressor of organ development, as did *CmWIP1* in carpel primordia of male melon flowers (Martin *et al.*, 2009). These data also demonstrated that WIP proteins could be recruited to inhibit growth of both vegetative and reproductive organs. >>

To take in consideration the reviewer comment we have moderated our conclusion as follow:

<< In conclusion, our results demonstrate that AtTT1/WIP1 transcription factor **could** act as strong suppressor of organ development, as did *CmWIP1* in carpel primordia of male melon flowers (Martin *et al.*, 2009). These data also **suggest** that WIP proteins could be recruited to inhibit growth of both vegetative and reproductive organs. >>.

Comment 10: Line 435 – 439 First of all, you should specify which plant you work with. Can you indicate in this paragraph that it is Arabidopsis?

As requested by the reviewer we added the following sentence: <<Arabidopsis (*Arabidopsis thaliana*) genotypes (accession *Columbia-0*) were grown under a 16-h-light/8-h-dark condition at 22°C in a growth room>> and edited the paragraph for more clarity.

Comment 11: Line 440 – The title of this: “Arabidopsis Plasmid construction and transformation” better “Plasmid construction and Arabidopsis transformation”.

As suggested by the reviewer, we changed the title.

Comment 12: Line 456 - “a the” too much of something.

We have disadvantaged this sentence...

Comment 13: Line 488 – 500 Have any biological or technical replicates been used in the CHIP-seq and CHIP-PCR analyzes?

See comment 2 for the replicates and the number of reads and % of mapped reads can be obtained in Figure S6. For more clarity we have indicated the number of replicates in M&M. Furthermore ChIP-seq of NTT and TT1 identified overlapping binding sites (see Figure 3C and D). For ChIP-PCR we have used 3 technical replicates (qPCR reactions) from separately collected material. Not the same that used for ChIP-seq.

Comment 14: Line 502 – Please provide information about IGB. A description has been included on M&M.

Comment 15: Line 504 - Please provide information about TSS. A description has been included on M&M.

Comment 16: Line 509 – 519 Have any biological or technical replicates been used in the RNA-seq analyzes?

We have generated two biological replicates per condition, *Dex:NTT* and *Dex:TT1* samples and their corresponding controls (*Dex:GR*). In total 8 RNA-seq libraries were deposited. For more clarity we have indicated the number of replicates in M&M. See also Comment 2 and Figure S6 and S7.

Comment 17: Line 535 – 537 Have any biological or technical replicates been used in the Quantitative PCR analyzes? What about statistics in this case?

RT-qPCR results from Figure 2B been obtained using the same RNAs than those for RNA-seq. We have used two biological replicates and three technical replicates per condition. Histograms from Figure S3C, S5D and Figure S9B have been obtained from the average of three technical replicates per genotype/condition.

Comment 18: Line 538 - Is the sequential data obtained by you available e.g. in the SRA or other databases?

The raw RNA-seq and ChIP-seq data described in this study have been deposited to the NCBI Short Read Archive (SRA) database under the BioProject ID PRJNA608903.

Comment 19: Figure 6 - Do these interactions mean something? Is this just a simple graphic? This is unclear and misleading.

We performed Gene Ontology (GO) analysis, using clusterProfiler, on shared DEG genes that were also bound by TT1 and NTT. Two main gene networks related to response to JA, ET and SA, and regulation of cell death were identified activated. These gene networks were also enriched with genes harboring W-box motif in their promoters. Similarly, meristem growth network and the network response to BR were both repressed. Their corresponding genes were also found harboring GAGA motif. We integrated all the information in the model in Figure 6. For more clarity we simplified the model.

Reviewer #2 (Remarks to the Author):

Comment1: This manuscript wants to clarify the molecular background of carpel arrest regulated by WIP1 gene and hypothesizes that WIP genes show conserved function across species. Large volume of transgenic work which benefit to understand the gene function was contained in this work. With these OE lines, the authors analysis the genome-wide WIP binding site and clarify the related pathways. The results show that WIP TFs repress development and growth by inducing plant defense response via hormonal variations.

In general, the manuscript reports interesting findings. The text is easy to follow and most of the figures are clear. Here, I have some questions:

In the name of the authors I would like to thank Reviewer 2 for acknowledging the importance of this work and his constructive suggestions.

Comment2: As we all know, the ethylene pathway together with the WIP1 gene decide the sex expression in Cucurbitaceae, but the detailed mechanism remains to be elucidated. Based on your data, could you provide some explanations? Or the mechanism is different in Arabidopsis?

In cucurbits the ethylene produced in the phloem companion cell of the flowers of the young vines controls the expression of the WIP; and the ethylene produced in the carpel control the development of the stamina. To investigate the function of the WIP proteins in Arabidopsis we disconnected the control of the expression of the WIP protein of the Ethylene, and using the DEX system to drive WIP expression. We did that to bring new insight to the function of the WIP proteins independently of the ethylene.

To take in consideration this important comment by the reviewer and for more clarity we developed further this point in the discussion.

Comment3: The gene expression patterns of those WIP members should be added.

As requested by the reviewer, gene expression analysis of the six WIP genes from Arabidopsis had been included on supplementary data (Figure S2).

Comment4: In melon and watermelon, the WIP1 is an apical shoot-specific gene, and the expression of WIP1 is detected in the flower buds at stages 1 to 5 before they diverge into the male or female organ. As shown in your work, the CRC promoter was expressed at stage 6 in carpel. Maybe you can add another earlier expressed promoter to compare the phenotype.

Sex determination in melon occurs at stage 4 to 6. In Arabidopsis CRC is expressed at stage 6. The choice of CRC is justified because it a very early gene required for carpel development without affecting the development of other organs in the plant. Our analysis using *AP3*, a B class floral homeotic gene specifically expressed in the petal and stamens, *GLABRA-1* and the *SOLITARY ROOTS* promoters further demonstrated that AtTT1/WIP1 transcription factor could act as general strong suppressor of organ development for both vegetative and reproductive organs.

Comment5: In your schematic model, WIP has a direct interaction with DNA, but there is no data to support. And how to explanation both WRKY and GAGA motifs can bind to WIP?

ChIP-seq analysis in general has the advantage to show genome wide interaction between the TF and the DNA in vivo. However it does not give information whether the interaction is direct or indirect. In Our analysis the produced ChIP-seq data clearly show a binding and recognition of specific domains by AtWIP1/TT1 and AtWIP2/NTT either directly or indirectly, via interaction with other DNA binding proteins that may specify recognition of WRKY or GAGA motifs. For more clarity we simplified the model.

Comment6: Minor points about the figures : In Fig.1A, why and how to divide the phylogenetic tree into two part (pink and blue background)?

We did that because NTT, WIP4 and WIP5, in one side and TT1, WIP3 and WIP6 in the other side are more phylogenetically related, respectively. Furthermore Crawford et al. (2015) have shown that NTT, WIP4 and WIP5 share redundant function in root development. Still for more clarity we have modified the phylogenetic tree, leaving the cluster of NTT, WIP4 and WIP5 only.

2. In Fig.1D, two transgenic lines (#6, #11) were shown. But in B and C, only one line, and which one?

For more clarity, we specifically indicated the name of the transgenic line (#6) we used in the figure.

3. In Fig 3, it is hard to understand that the target gene of WIP is more than 6000. Any explanation?

This not an unusual number of binding sites of a transcription factor. Many examples in literature show similar number of bindings, here few examples.

- ChIP-seq analysis for three B-ARRs (5128 targets for ARR1, 6272 targets for ARR10, and 6240 targets for ARR12). Xie et al. Nature Communications Vol 9, 1604 (2018);
- Subo et al. Identified 4,861 binding sites for ARR10. PNAS July 18, 2017 114 (29) E5995-E6004; <https://doi.org/10.1073/pnas.1620749114>;
- Theune et al. 5873 BPC6-target regions were identified. Front. Plant Sci., 16 April 2019 | <https://doi.org/10.3389/fpls.2019.00466>

Reviewer #3 (Remarks to the Author):

Comment1: In this report, Roldan et al. unveiled a hidden commonality in the function of WIP genes, using Arabidopsis as a model. The authors started from the mystery that CmWIP1, which can act for sex determination in melon, is nested into the gene categories with the functions relating seed color or fertilization of silique in Arabidopsis. They showed that the fundamental protein functions of these WIP family genes are almost identical, where overexpression can act for repression of the growth and senescence of leaves, with a few exceptions. Combination of ChIP-Seq and transcriptomic data suggested the same molecular function of these WIPs to recognize the common cis-motif and to regulate similar pathways. Lastly, the authors showed the possibility that these WIPs probably can work for organ-specific repression depending on the promoters, which might explain the sex determining function in Cucumis. The methods sounded, and the conclusion was convincing and very clear. I really enjoyed reading this manuscript, although I would raise some suggestions and questions on the contents of this manuscript, as followings.

In the name of the authors I would like to thank Reviewer 3 for his statement: << The methods sounded, and the conclusion was convincing and very clear. I really enjoyed reading this manuscript.>>

Comment2: CmWIP1 would have a very important position to explain plant sexuality based on ethylene signaling. Thus, if the authors avoid the discussion about CmWIP1, the manuscript might be only one of the many papers involving Arabidopsis transcription factors. However, although the authors started from CmWIP1 in melon, the discussion (not the results) was focused only in Arabidopsis situations. I strongly recommend to write about the (estimated) regulatory systems of CmWIP1 via ethylene signaling (or ACS regulation) to determine sexuality in melon. I guess that the part on the WRKY motif (Figure 4) and the last part (Figure 5) might be directly related to the discussion of sex determination in melon. CmWIP1 can recognize W-box in ACS-7 in androecium/gynoecium specific manners?

As suggested with the reviewer we discussed further how the investigation of WIP in Arabidopsis is helping us to understand the sex determination in melon.

Comment3: TT1-induced Arabidopsis under the control of CRC- or AP3-promoters showed clear conversion into male and female flowers. Partially related to the question above, were their regulatory networks similar to melon's male and female flowers? If the authors concept that WIP genes potentially carry identical trans-functions (and the function differentiation may be dependent on cis-regulation), these sex-converted Arabidopsis would be able to reflect the expression differentiation between male/female melon.

Yes this is exactly our conclusion. Cucurbits are using WIP protein to inhibit the carpel and produce unisexual male flowers. To inhibit stamina they are using an independent pathway, the Ethylene pathway. To produce male and female flowers on the same plant cucurbits use ethylene to control the expression of the WIP protein. All this is now further discussed in the paper.

Comment4: Minor points

1. Figure 1A: Please add outgroup gene and genes from other species, for definition of the distinct evolutionary clades. The current clustering looks very arbitrary.

We have addressed this question and generated a new phylogenetic tree including an outgroup gene, ATSTOP1, a gene that encodes for a nuclear Cys(2)His(2)-type zinc finger protein involved in H⁺ and Al³⁺ rhizotoxicity. We have changed the background, only the cluster of NTT, WIP4 and WIP5 for which a redundant function has been described in the literature (Crawford et al. 2015) has been highlighted.

2. In some pictures, such as Figure 2C-D, 5A-C, and 5G-K, scale bars missed.

Scale bars were added

3. Figure 4F-G: quite confusing figures. Please add X- and Y-axis scales. For my viewer, bars in the peaks are missing. The authors should add more interpretations of these panels.

We have included more information in the legend to add information about those figures.

REVIEWERS' COMMENTS:

Reviewer #1 (Remarks to the Author):

The authors answered my questions comprehensively. The corrections made in the manuscript are fully sufficient.

Reviewer #2 (Remarks to the Author):

The revised MS is more clarify and acceptable.

Reviewer #3 (Remarks to the Author):

I have confirmed that the authors kindly have reflected all of my comments and suggestions.
Thanks,